# Whale-Associated Microbial Communities Remain Remarkably Stable despite Massive Water Community Disruption in a Managed Artificial Marine Environment

William Van Bonn [1,*], Francis Oliaro [1] and Lee Pinnell [2]

1 Animal Care and Science Division, John G. Shedd Aquarium, Chicago, IL 60605, USA; foliaro@sheddaquarium.org
2 Veterinary Education, Research, and Outreach Program, Texas A & M University, Canyon, TX 79015, USA; ljpinnell@cvm.tamu.edu
* Correspondence: bvanbonn@sheddaquarium.org

**Abstract:** Highly managed and built environments such as zoos and aquaria provide a rich source of standardized environmental monitoring data over periods of years to decades. A fifty percent water change in an 11.4-million-liter indoor artificial sea water system housing three species of marine mammals was conducted over a two-month period. Using 16S rRNA gene sequencing, the microbial community structure of the system water and three host sites (feces, skin, and exhaled breath "chuff") of whales housed in the system were characterized. Diversity measures confirmed massive disruption to the water community structure as an expected result of the water change. Host site-associated communities remained remarkably stable. Improved understanding of host microbial community dynamics in response to environmental system perturbations allows for sound management decisions toward optimizing conditions for resident animals.

**Keywords:** aquatic microbial ecology; cetacean; aquarium; microbiome

## 1. Introduction

The assembly of microbial communities associated with host animals and the environment in which the animals live is subject to multiple abiotic and biotic drivers. Host-associated and environmental microbes may be sources of infection leading to animal diseases; however, they may also serve to exclude or compete with potential pathogens and reduce disease incidence rates. Our understanding of the role host-associated microbes play in host health is markedly improving. For example, beneficial bacteria have been identified that protect amphibian skin from invasion by the fungal pathogen *Batrachochytrium dendrobatidis* [1]. There is a paucity of similar work with Cetaceans.

Animals of the order Cetacea (whales and dolphins) are highly charismatic mammals physiologically adapted to an obligate aquatic existence. These animals draw much attention in public display facilities (ex situ) and as subjects of scientific studies in their native habitats (in situ). A growing number of such studies have investigated microbial communities associated with multiple body sites from several species of cetaceans [2–7]. Skin, feces, oral, and respiratory sourced samples ("exhaled breath condensates", "chuff" or "blow") are the most frequently reported animal-associated sample sites. Cetaceans breath hold after inspiration; that is, they surface, exhale, inhale and dive. The surface exhalation of free-ranging animals tends to be forceful and in larger cetaceans is relatively easily sampled by positioning a capture device directly over the external nare(s) known as the blowhole(s). Several investigators have utilized unmanned aerial vehicles to collect samples from free-ranging whales [8,9] while others have utilized petri dishes attached to poles maneuvered from a surface vessel [10]. Under direct human care in aquaria, whales and dolphins can be operantly conditioned to forcefully exhale in response to a stimulus

facilitating regular sample collection. These animals are also typically conditioned to allow collection of feces and surface swabs for microbiological analyses. Despite the ease of collection of samples from animals in controlled environments, most cetacean microbiome studies to date have been surveys of animals in natural environments describing community characteristics by body site, species, or individual with some geographic and temporal variation reported. Others have been conducted in controlled systems that use natural sea water as source water.

Built aquatic environments that rely on recirculation of water require periodic water changes to address the buildup of compounds such as nitrate nitrogen that is not addressed by the life support system. The John G. Shedd Aquarium manages an 11.4-million-liter indoor completely artificial marine system, known as the Abbott Oceanarium, that has housed cetaceans for over three decades. An investigation of dolphin-associated microbiota and interactions with environmental source microbes was conducted using this system and resident animals [11]. That study demonstrated that continuous microbial exposure occurred between all sites, yet each environment maintained a characteristic microbiota, suggesting that the majority of exposure events do not result in colonization. Participating animals received probiotic supplements during the study period, but no environmental parameter values were intentionally manipulated. Small changes in water physiochemistry had a significant but weak correlation with change in dolphin-associated bacterial richness but had no influence on phylogenetic diversity [11]. Changes in environmental microbial community structure in response to system disturbances have been described in experimental aquariums [12] and our group used a smaller artificial saltwater system at Shedd to document aquarium microbiome response to a 90% water change [13]. In the study reported herein, a planned fifty-percent water change in the oceanarium system was conducted by an initial dilution with fresh water, followed by salt additions. This provided an opportunity to monitor environmental and whale host-associated microbial communities from three body-sites for change associated with the system disturbance. To our knowledge, this is the first description of the response of host-associated microbial communities of a cetacean to a planned environmental disruption. It provides valuable insight into how these large obligate aquatic mammals manage microbial exposure in a changing environment.

## 2. Materials and Methods

### 2.1. Environmental Samples

The study was conducted over four phases from 8 January through 26 February 2020. During Phase One (pre), from 8–18 January, no manipulation was done to the system and baseline samples were collected. Over Phase Two (transition), from 19 to 26 January, fresh water was continuously added to the system resulting in a salinity drop from 31 ppt to 20 ppt. A commercial salt mix (Instant Ocean®, Blacksburg, VA, USA) was then added to the water to bring the salinity to approximately 24 ppt where it was held through 15 February as Phase Three (hold). During the last phase, Phase Four (post), salt was added until salinity was approximately 29 ppt and samples were collected until 26 February. To increase sample size and facilitate statistical comparisons, continuous data representing samples collected at each date were grouped into these four phases and analyzed as categorical variables.

Three one-liter replicates of water were collected from the habitat daily into sterile Nalgene® bottles. Each replicate was then pulled through a sterile 0.2 micron pore size filter membrane (Pall®, Port Washington, NY, USA) using a Vacuubrand® vacuum system and manifold. After filtering, membranes were placed in 5 mL PowerBead tubes (QIAGEN, Germantown, MD, USA) and stored at −70 °C prior to DNA extraction.

### 2.2. Whale Samples

Samples from three sites were collected from each of six resident beluga whales (*Delphinapterus leucas*) once per week during the study period. Three replicates were collected at each site each time the site was sampled. Skin samples were obtained by swabbing in the

axilla and/or pectoral appendage held above the water surface. Exhaled breath condensate (chuff) was collected by holding a sterile inverted petri dish approximately 10 cm directly above the animal's blowhole and asking the animal to exhale forcefully. The petri dish was then immediately swabbed with three separate swabs. Feces from each whale was collected by gently inserting a sterile 15 French red rubber tube per anus to a depth of approximately 15 cm, immediately withdrawing and swabbing feces from the tube surface and lumen. All swabs were stored in WhirlPak® (Nasco, Madison, WI, USA) bags at −70 °C prior to DNA extraction. Medical and husbandry records were reviewed for all participating animals throughout the study period. All participating whales remained clinically normal. All animals' appetites, attitudes and cooperative behaviors remained unchanged. None were receiving any medications and all routine surveillance hematology and chemistry values were within expected values for this species for the duration of the study period.

### 2.3. Sequencing

DNA was extracted from water filters using the MagAttract Power Water DNA/RNA kit (QIAGEN) and from animal samples using the MagAttract Power Soil DNA KF kit (QIAGEN) following the manufacturer's instructions. Extractions were carried out using the KingFisherÔ Flex Purfication System (Thermo Fisher, Waltham, MA, USA). Blank field and lab control samples were processed with each batch of extractions consisting of sterile filters or swabs (field control) and empty wells containing only extraction reagents (lab control). Control blanks were taken through library prep and sequencing steps described herein.

Bacterial and archaeal DNA was amplified using primer constructs (515 f/806 rB) targeting the V4 region of the 16S rRNA gene [14]. The constructs contain Illumina specific adapters followed by 12 bp Golay barcodes on each forward primer, primer pads and linkers as well as the template specific PCR primer at the 3′ end. PCR was performed in replicate 25 µL reactions containing 12.5 µL Phusion Hot-Start Flex 2× MasterMix (New England Biolabs, Ipswich, MA, USA), 0.2 µM final concentrations of forward primer 515 f and reverse primer 806 rB, 2 µL of template DNA and nuclease free water to equal 25 µL. Mock microbial community DNA standards (Zymo Research, Irvine, CA, USA) and negative controls containing no template DNA were prepared with each PCR replicate. Thermal cycling conditions were carried out as follows: 98 °C for 30 s, 30 cycles at 98 °C for 10 s, 55 °C for 30 s and 72 °C for 30 s, with a final extension of 5 min at 72 °C. Replicate PCR products were combined and 5 µL of each was electrophoresed in 1.8% agarose gels to confirm amplification of the V4 region. An amount of 25 µL of each PCR product was cleaned and normalized using the SequalPrep™ Normalization Plate Kit (Applied Biosystems, Waltham, MA, USA) and equal volumes were pooled together to create a normalized library. The pooled amplicon library was quantified using a Qubit™ 3.0 fluorometer and Qubit™ dsDNA HS Assay Kit (Life Technologies, Waltham, MA, USA). The molarity of the pooled library was calculated and diluted to a loading concentration of 8 pM with addition of 10% PhiX Control library (Illumina, San Diego, CA, USA) to increase sequence diversity. Paired-end sequencing for a total of five hundred cycles was conducted on two Illumina MiSeq runs using custom sequencing primers described previously [15].

### 2.4. Bioinformatics

Demultiplexed paired-end reads were imported into QIIME2 version 2019.7 [16] and processed as described previously [17]. Forward reads were truncated at 237 bp and trimmed from 2 bp, while reverse reads were truncated at 214 bp and trimmed from 7 bp. Taxonomy was assigned using a Näive Bayes classifier trained on the SILVA 132 99% OTUs database, where sequences had been trimmed to only include those base pairs from the V4 region bound by the 515 f/806 r primer pair. A midpoint-rooted tree was generated under default settings, which was used for phylogeny-based metrics.

Data were then imported into phyloseq version 1.26.1 [18] and samples with less than 1000 amplicon sequence variants (ASVs) were discarded. Richness, Shannon's diversity and Faith's phylogenetic distance for the remaining samples were calculated using the

'estimate_richness' function within phyloseq and the 'estimate_pd' function within the package btools. Differences in alpha-diversity were tested using pairwise Wilcoxon rank-sum tests with a Benjamini–Hochberg correction for multiple comparisons.

Samples were then normalized with total sum scaling using the 'transform_sample_counts' function and the lowest remaining ASV count of 1051 to account for differences in sequencing depth. Generalized UniFrac distances were then calculated using the 'GUniFrac' package [19], and a principal coordinates analysis (PCoA) was calculated and plotted from these distances using the 'ordinate' and 'plot_ordination' functions in phyloseq. A permutational multivariate analysis of variance (PERMANOVA) with a Benjamini–Hochberg correction for multiple comparisons and 999 permutations was used to test for significant differences in microbial community composition using the 'vegan' [20] and 'pairwiseAdonis' [21] packages in R version 3.6.1 [22]. To ensure differences in microbial communities were not due to unequal dispersion of variability among groups, permutational analyses of dispersion (PERMDISP) using 999 permutations were conducted for all significant PERMANOVA outcomes with the 'vegan' package in R.

Further, the relative abundances of amplicon sequence variants (ASVs) within each sample were calculated and plotted. Unless specified otherwise, R version 3.6.1 [22] was used for statistical analysis of data. Shapiro–Wilk tests were used to test data for normality. Differences in relative abundance values were tested using pairwise Wilcoxon rank-sum tests with a Benjamini–Hochberg correction for multiple comparisons.

## 3. Results

### 3.1. All Sources

Richness (observed ASVs), Shannon (diversity), and Faith's phylogenetic diversity for all sample sources across all time points are visualized as boxplots in Figure 1. Except for the higher diversity of chuff communities, the richness, diversity, and phylogenetic were all significantly higher within water communities than all whale-associated communities (Figure 1, pairwise Wilcoxon rank-sum test with Benjamini–Hochberg correction, $n$ = 130–149, $p < 0.05$). Among whale-associated communities, richness and diversity were also significantly higher in chuff communities than skin and fecal communities, and skin communities were also significantly richer and more diverse than fecal communities (Figure 1, pairwise Wilcoxon rank-sum test with Benjamini–Hochberg correction, $n$ = 130–149, $p < 0.05$). However, phylogenetic diversity was similar between skin and chuff communities, which were both more phylogenetically diverse than fecal communities (Figure 1, pairwise Wilcoxon rank-sum test with Benjamini–Hochberg correction, $n$ = 130–149, $p < 0.05$). Of the total 8801 ASVs across all samples, 1462 were shared between water and whale communities, while 4821 were unique to water and 2518 were unique to whale-associated communities (Figure S1A). Of the 2518 ASVs unique to whale-associated communities, more than two-thirds were from skin samples (~68%, Figure S1B). Only 108 of the 8801 total ASVs were shared between water and all three body sites (skin, chuff, feces) and the largest number of shared ASVs was between water and skin samples (1186 ASVs, Figure S1B).

Differences in community composition were compared using generalized, UniFrac distances, principal co-ordinates analysis (PCoA), and PERMANOVA. Across all time points, water communities were significantly different from all three whale-associated communities (skin, chuff, feces), which were also significantly different from each other (Table S1; PERMANOVA with Benjamini–Hochberg correction, $n$ = 130–149, $p < 0.05$). Visualization with PCoA confirmed that all four sample types were structured different from each other (Figure 2), despite significantly higher and lower dispersions of variances in skin and fecal samples, respectively (Table S1; PERMDISP, $n$ = 130–149, $p < 0.05$). Skin samples were more varied than other body sites (Table S1) and were more similar to the environment (Figure 2; Table S1).

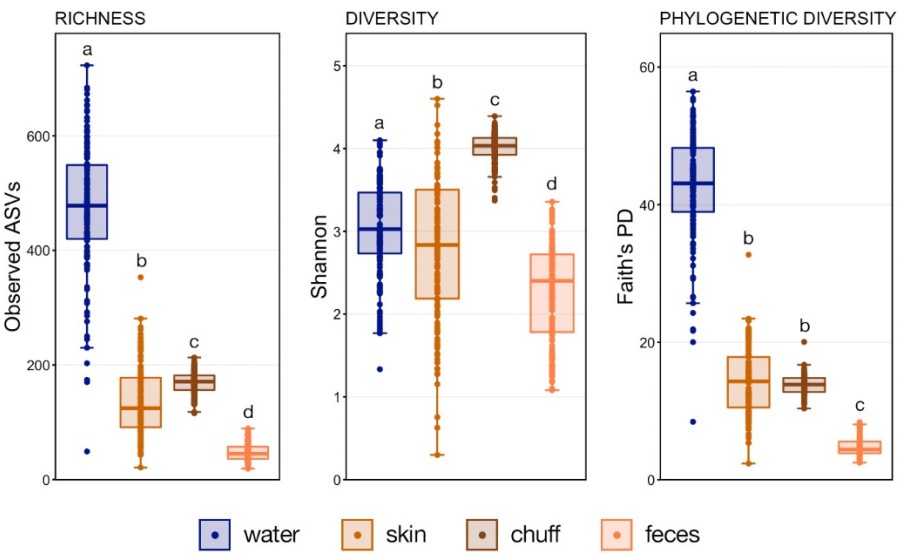

**Figure 1.** Boxplots demonstrating differences in richness (Observed ASVs), diversity (Shannon), and phylogenetic diversity (Faith's PD) between water (*n* = 149), skin (*n* = 130), chuff (*n* = 142), and fecal (*n* = 144) samples. Significant differences are illustrated by different letters (pairwise Wilcoxon rank-sum with Benjamini–Hochberg correction, *n* = 130–149, *p* < 0.05).

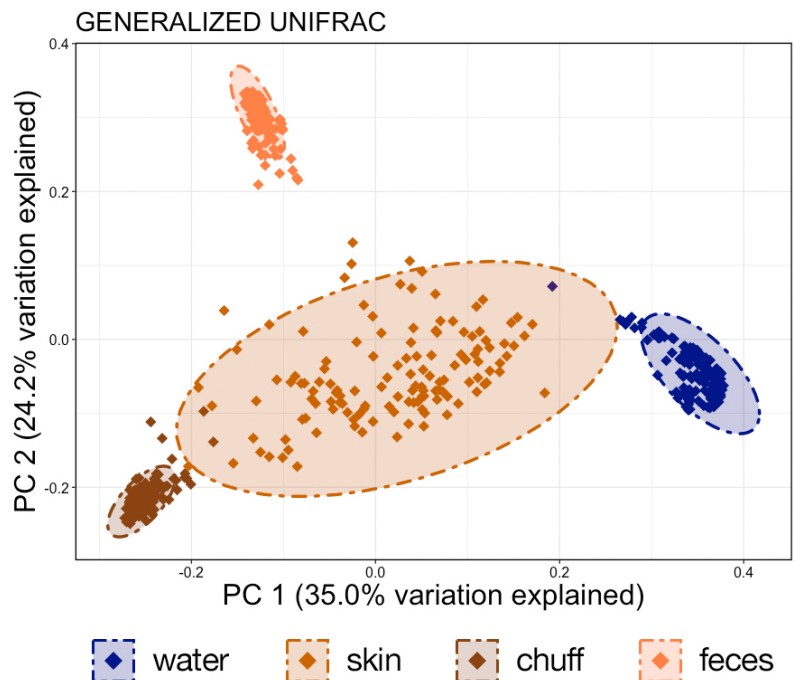

**Figure 2.** Principal co-ordinates analysis (PCoA) of generalized UniFrac distances illustrating differences in microbial community composition between all samples across all timepoints. The PCoA demonstrates clustering of 16S rRNA gene sequences from water (*n* = 149), skin (*n* = 130), chuff (*n* = 142), and fecal (*n* = 144) microbial communities. Ellipses represent 99% confidence ellipses for each sample type.

*3.2. Water*

Changes in the salinity and α-diversity values of the water in the system over the study period are shown in Figure 3. The phases of the study delineated by the changes in the measured salinity over the four time periods are clearly seen. Also clearly visible is the massive disruption in microbial community diversity during Phases Two and Three. Notably, there is a decrease in richness and diversity as salinity levels drop toward their

lowest values. During Phase Three, as salinity is increased and held near 24 ppt, richness and diversity rebound. Mean diversity values are significantly different between the pre and post water change phases (pairwise Wilcoxon rank-sum, $p < 0.05$), indicating a more diverse microbial community post water change.

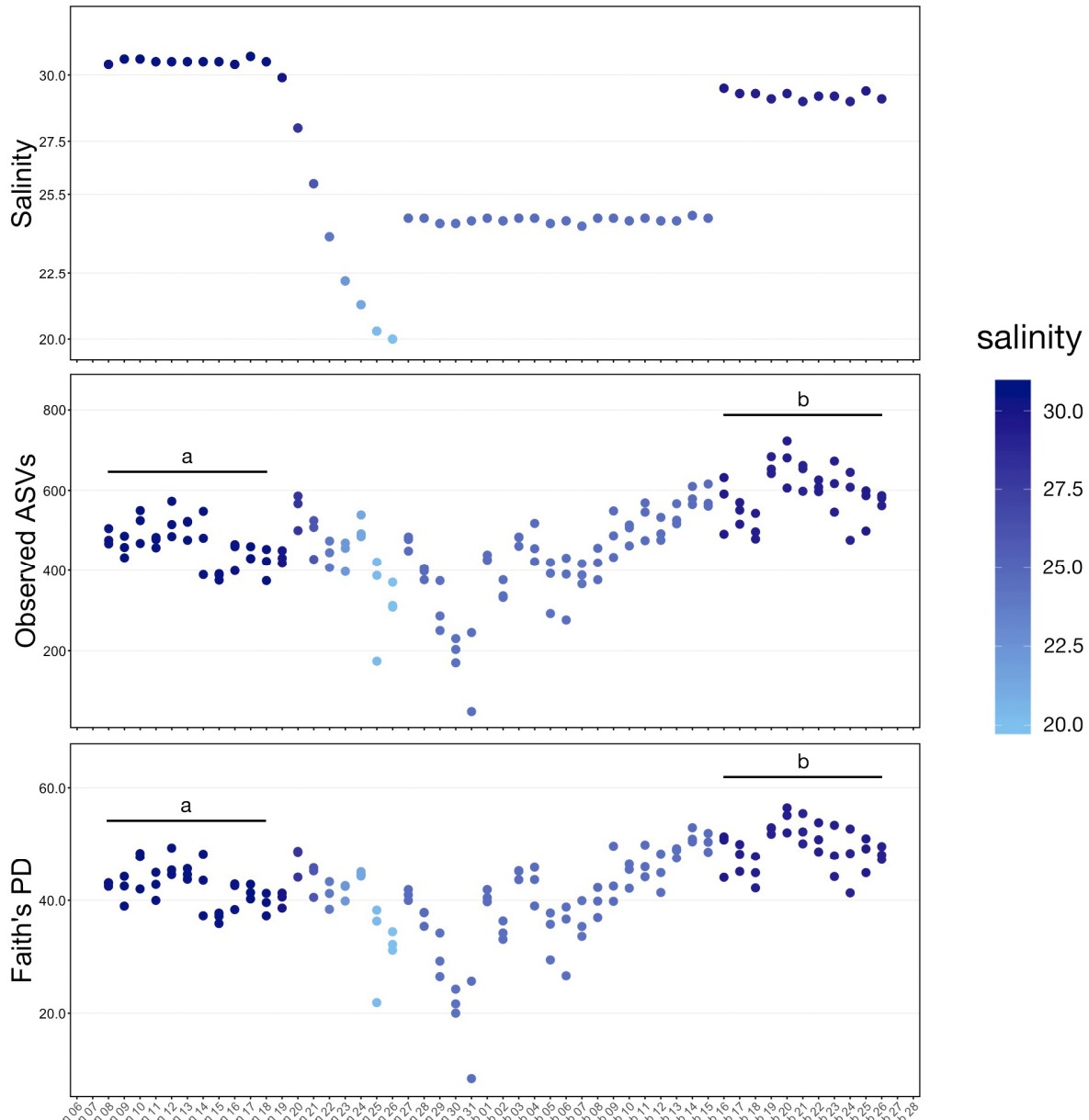

**Figure 3.** Scatterplot tracking changes on salinity (**top**), richness (Observed ASVs—**middle**), and diversity (Faith's PD—**bottom**) in water communities as the water change progressed from 8 January to 26 February. Points are colored based on salinity values. Significant differences in mean diversity values between pre- (8–18 January; *n* = 33) and post- (16–26 February; *n* = 33) water change are illustrated by different letters (Kruskal–Wallis, $p < 0.05$).

Figure 4 is a PCoA plot of all water samples to illustrate β-diversity (generalized UniFrac distance) by phase. Samples are colored by phase (pre = Phase One, transition = Phase Two, hold = Phase Three, post = Phase Four). All phases were significantly different from each other (PERMANOVA with Benjamini–Hochberg correction, *n* = 33–59, $p < 0.05$). Communities were more varied during the transition (Phase Two) and hold (Phase Three) phases compared with pre (Phase One) and post (Phase Four) phases. The mean relative abun-

dance of bacterial families making up greater than 1% of the overall water community for each sample is visualized in a stacked bar plot by date for the duration of the study period (Figure 5). Notable striking changes are the loss of the Arenicellaceae (Phase One, mean relative abundance = 15.69%, SEM 0.495; Phase Four, mean relative abundance = 1.13%, SEM 0.067) which occurred toward the end of Phase Two (transition) when salinity decreased and the increase in Nitrosopumilaceae (Phase One, mean relative abundance = 4.94%, SEM 0.191; Phase Four, mean relative abundance = 11.34%, SEM 0.245) as salinity was increased and held in Phase Three (hold) and further increased in Phase Four (post). Mean relative abundance values for the top twenty-five most abundant taxa are reported in Table S2.

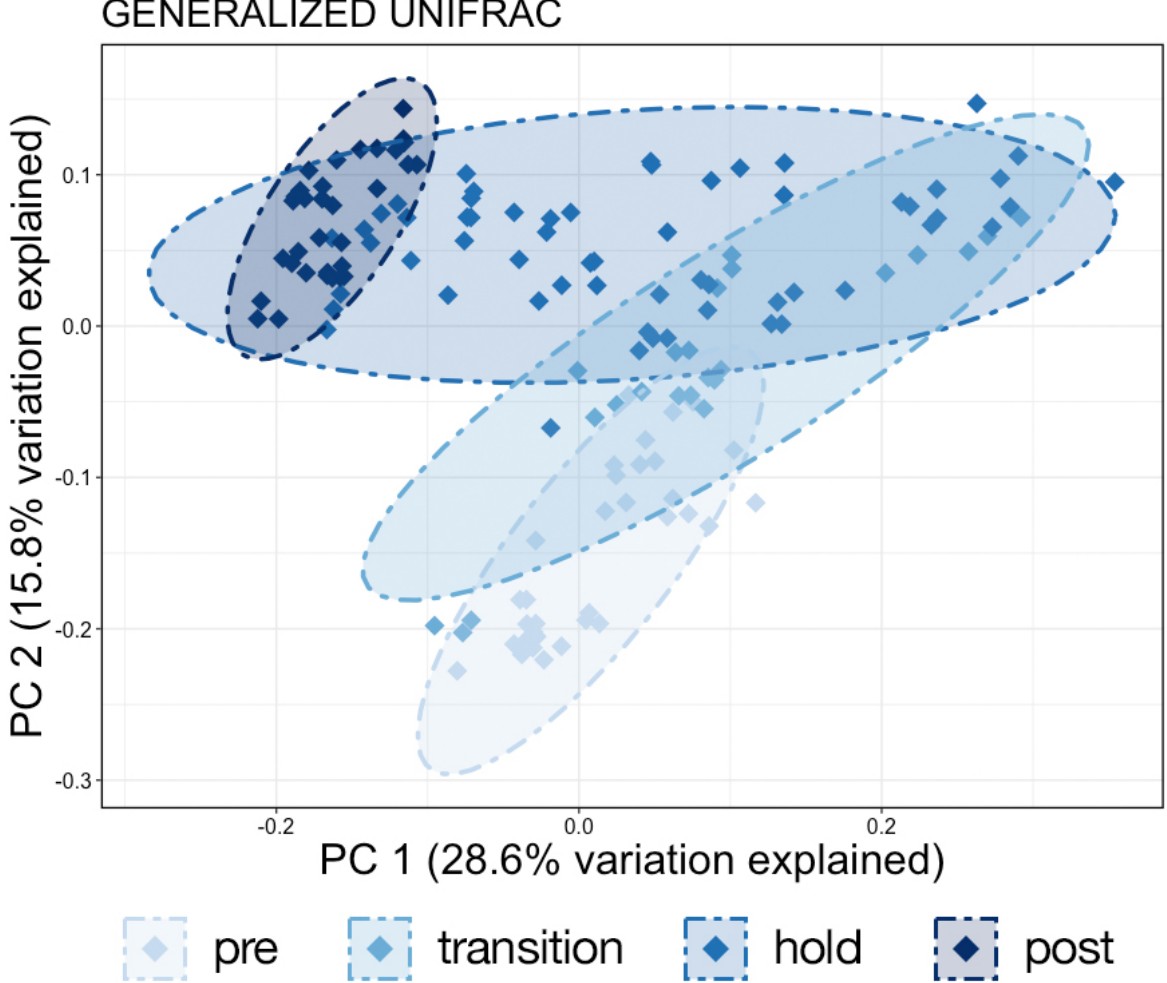

**Figure 4.** PCoA of the microbial communities from water samples throughout the water change. To aid visualization, samples are colored by phase (pre *n* = 33, transition *n* = 42, hold *n* = 59, post *n* = 33) opposed to salinity. All phases were significantly different from each other (PERMANOVA with Benjamini–Hochberg correction, *n* = 33–59, *p* < 0.05).

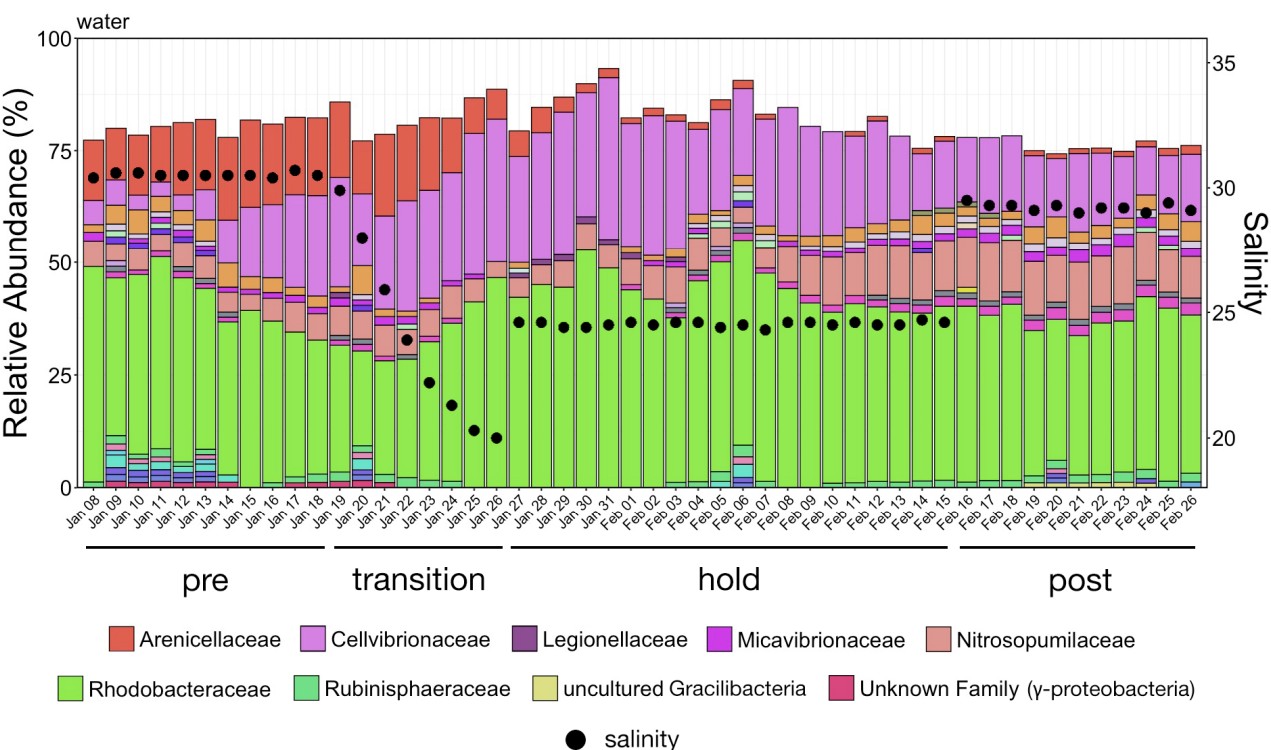

**Figure 5.** Barplot demonstrating the mean relative abundances of families making up greater than 1% of the overall community for water samples throughout the water change. Salinities for each day are overlayed with black points. The 9 most abundant families across are samples are displayed in the legend.

### 3.3. Whales

Scatterplots of richness (Observed ASVs) and Faith's PD for every skin sample collected and the water salinity throughout the study period are presented in Figure 6 (pairwise Wilcoxon rank-sum test with Benjamini–Hochberg correction). Despite significant changes to α-diversity of the water community (Figure 3), there were no significant changes in α-diversity of skin communities over time (Figure 6). There were no significant changes in α-diversity or β-diversity measures for fecal (Figures S2 and S6) or chuff (Figures S3 and S7) communities over the study period. Significant changes in skin β-diversity were detected between phases, with the exception of the hold versus post phase (Figure 7), although differences are subtle compared with changes in water community structure. A bar plot of the mean relative abundance of bacterial families in skin swab samples over time (Figure 8) and mean relative abundance values for each phase of the study (Table S3) demonstrate the relative stability of dominant families at this site. Mean relative abundances of the two most abundant families Rhodobacteraceae (Phase One, mean relative abundance = 26.84%, SEM 2.923; Phase Four, mean relative abundance = 23.43%, SEM 3.430) and Flavobacteriaceae (Phase One, mean relative abundance = 15.84%, SEM 2.241; Phase Four, mean relative abundance = 16.33%, SEM 2.797) changed little. Significant changes observed in skin β-diversity were likely driven by a decrease in the family Saccharospirillaceae (Phase One, mean relative abundance = 15.4%, SEM 3.795; Phase Four, mean relative abundance = 1.74%, SEM 0.466). Supplemental Figures and Tables S4 and S5 visualize these same data for chuff and fecal communities, respectively.

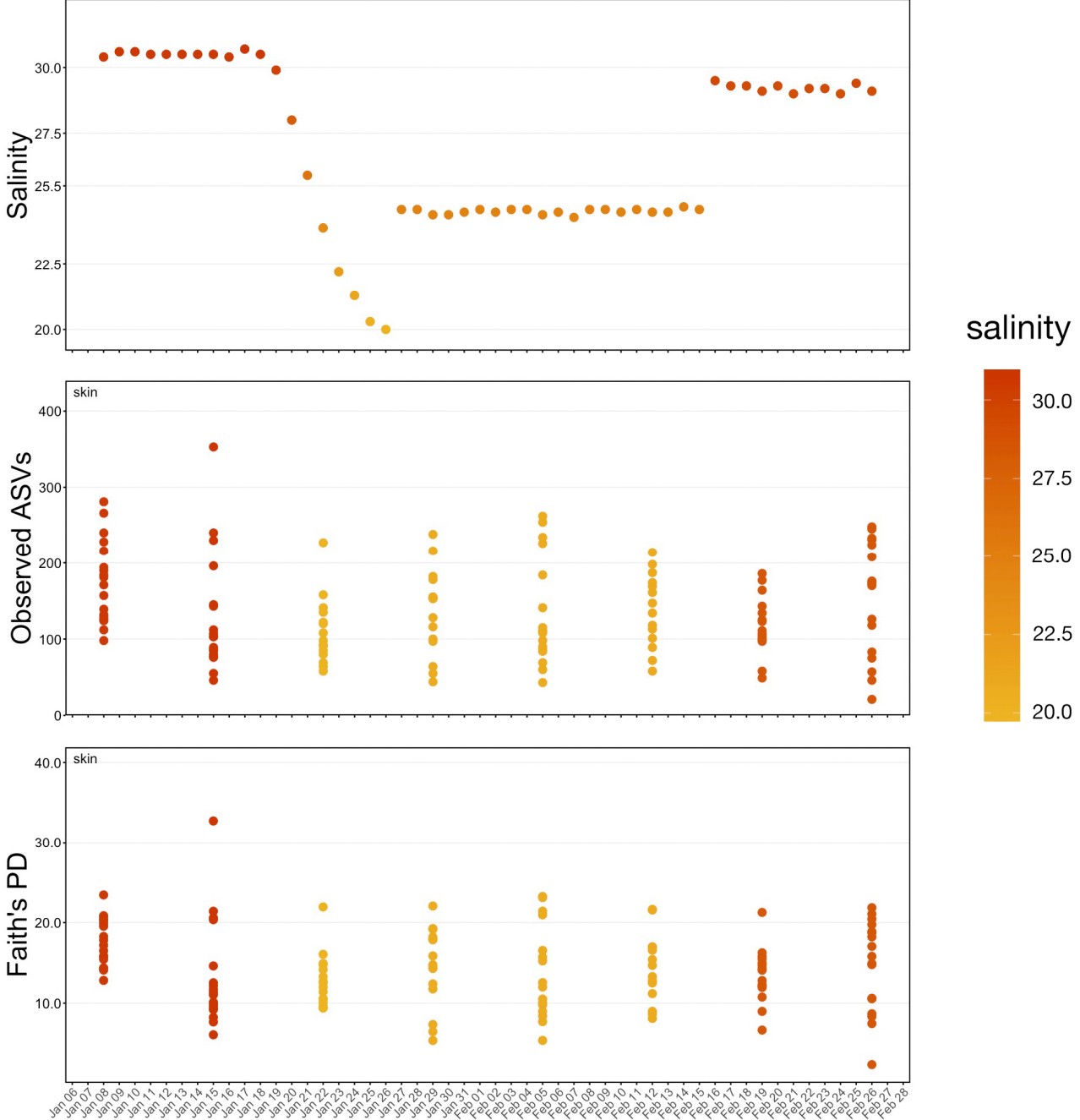

**Figure 6.** Scatterplot tracking changes on salinity (**top**), richness (Observed ASVs—**middle**), and diversity (Faith's PD—**bottom**) in skin community diversity. Points are colored based on salinity values. There were no significant changes in diversity.

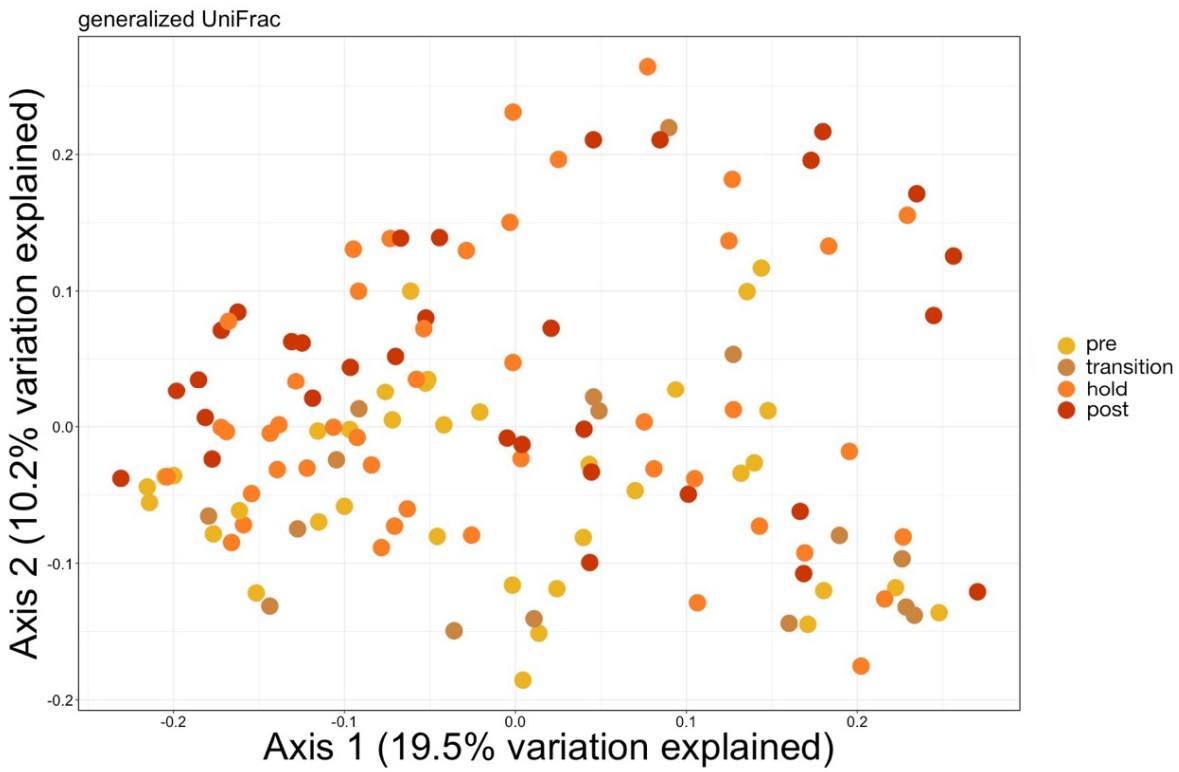

**Figure 7.** PCoA of the microbial communities from skin samples throughout the water change. To aid visualization, samples are colored by phase opposed to salinity. All phases were significantly difference from each other, with the exception of hold vs. post (PERMANOVA, $p < 0.05$).

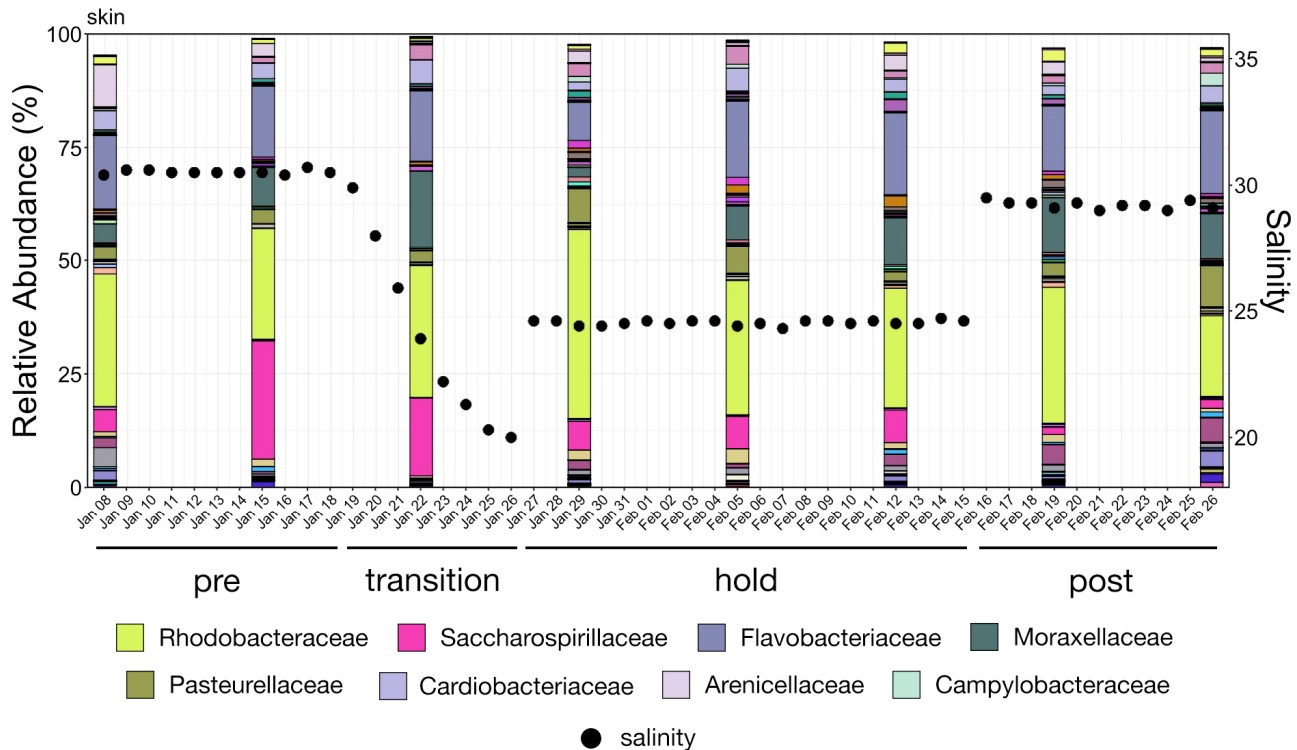

**Figure 8.** Barplot demonstrating the mean relative abundances of families within skin samples throughout the water change. Salinities for each day are overlayed with black points. The 8 most abundant families across are samples are displayed in the legend.

## 4. Discussion

Whale-associated microbial communities generally had lower richness, diversity, and phylogenetic diversity as compared to water communities as we previously reported in this same system [11]. Interestingly, the Shannon diversity of chuff communities was higher despite lower richness and phylogenetic diversity, suggesting that this increased diversity was limited to closely related taxonomic lineages. Furthermore, our results show massive disruption of the microbial community of the water as expected during the water change. However, the structure of whale-associated communities changed little (skin) or not at all (fecal, chuff) over the study period. Skin communities were the most varied and most similar to water communities, and although changes in skin β-diversity were found, they did not mimic shifts of abundant taxa observed in the water. This suggests some host-associated (endogenous) means of selective exclusion of some microbes and the establishment of a host 'core microbiome'.

Numerous studies describe microbial communities associated with skin surfaces, chuff, or feces of several species of cetaceans in both native environments and aquaria [2,4–7]. The skin-associated bacteria of free-ranging Humpback whales (*Megaptera novaeangliae*) from four geographically distinct areas were characterized from opportunistic biopsy or sloughed samples [2]. That study demonstrated that skin-associated microbial communities were less rich than surrounding sea water (lower absolute number of OTUs) as we observe in our data. Skin-associated organisms in the genera *Tenacibaculum* and *Psychrobacter* were prominently abundant in all Humpback whale samples. The authors suggest that these genera may be part of a core microbiome of the Humpback whale. These genera were also prominent in our whale skin samples, and were among the top four most abundant genera during each of the four water change phases (Table S3).

Van Cise et al. [7] describe the skin-associated microbiota of two populations of Alaskan free-ranging beluga also using biopsy obtained samples. They reported no taxa found across all individuals sampled and speculated that beluga may not have a species core microbiome. Unfortunately, they did not sample seawater. The definition of core microbiome is not universally agreed upon [23] and is likely to mature as further studies are conducted; however, skin-associated microbial communities are routinely found to be distinct from the surrounding seawater as is the case in our data. In the Alaskan group of belugas, *Physchrobacter* was not among the most abundant genera found; however, interestingly, it was one of two genera almost ten-fold more abundant in animals the authors characterized as diseased when compared to those they classified as healthy [7].

Another study evaluated the skin microbiota of French aquarium-held killer whales (*Orcinus orca*) and bottlenose dolphins (*Tursiops truncatus*) [4]. This group also reported significantly higher richness of the water than skin-associated communities and a dominance of the *Phsychrobacter* genus in skin communities. It is important to note that, although this was an aquarium environment, in contrast to our study system, the source water was natural seawater and there was no oxidation of the water as part of the treatment.

Marine mammals housed in zoos and aquariums in the United States are subject to standards in the Animal Welfare Act (AWA), promulgated by the United States Department of Agriculture (USDA), Animal and Plant Health Inspection Service (APHIS). These standards include water quality standards which state: The coliform bacteria count of the primary enclosure pool shall not exceed 1000 MPN (most probable number) per 100 mL of water. Should a coliform bacterial count exceed 1000 MPN, two subsequent samples may be taken at 48 h intervals and averaged with the first sample. If such an average count does not fall below 1000 MPN, then the pool water shall be deemed unsatisfactory, and the condition must be corrected immediately [24]. These standards are presumably based upon human recreational water exposure standards. There are no data supporting elevated total coliform counts as a specific health risk to marine mammals. Most facilities easily meet this standard by incorporating oxidation into the water treatment system for exhibits housing marine mammals. Oxidation puts significant selection pressure on the microbial communities of the water. We have previously shown that the β-diversity of the

water is the biggest difference between this system when compared to natural seawater systems [11]. This has held true for every system utilizing ozone as oxidant treatment that we have examined, including those using natural seawater as source water (Van Bonn, unpublished data).

Changes in the way waters are 'disinfected' may reduce the incidence of infectious disease in resident marine mammals [25] but much is yet unknown about how to optimize environmental conditions. Despite massive changes in the water microbial communities in response to a 50% water change, whale skin, chuff and fecal community structure remained stable. These results suggest that, during a short-term environmental challenge, host factors strongly influence which microbes can successfully colonize the three sites evaluated. However, we acknowledge that prolonged stress may change how microbial taxa respond.

In a survey of skin-associated bacterial communities of 89 Humpback whales, samples were collected from whales along the Western Antarctic Peninsula (WAP) early in the early austral summer in 2010 and late in the austral summer in 2013. The authors of that study defined a core skin microbiome for those animals and demonstrated a shift between the animals' early foraging season and the late foraging season. They also noted differences in whale skin microbiomes from whales in different geographic regions of the WAP. The authors speculated that the shift could have been due to nutritional factors as the animals moved from a catabolic state to an anabolic state or that temperature could have played a role. Water samples were not analyzed in that study, but the authors stated that because previous studies have shown cetacean skin communities differ significantly from local water it is unlikely that variation in the skin microbiome is due to alterations in the surrounding water [26]. Our findings support this speculation, despite massive community shifts in the water in which the animals were living, the skin, chuff and fecal communities were remarkably stable.

The water temperature of our study system is changed with season to approximate natural systems. Over the study period, the water temperature varied from 12.6 °C to 15.0 °C. Although this may have contributed to the change in water community structure, there is no evidence that it influenced skin-associated communities.

The mechanisms by which cetaceans selectively permit colonization by microbes have not been investigated and are likely to be complex, involving both innate and acquired immune functions. Our data show that these mechanisms are functional during acute large shifts in the water community as a result of 50% water change in the stable artificial saltwater system with no other environmental parameter value changes. Characterizing the immune system of a marine mammal and translating to fitness or disease resistance is extremely difficult [27]. Continued work needs to be conducted to further understand the role that microbes, such as those of the genera *Tenacibaculum* and *Psychrobacter*, may play in maintaining cetacean host health.

### 5. Conclusions

In summary, we documented stable host-associated microbial communities despite an acute major shift in the water community in an otherwise stable artificial sea water test environment. Our results provide further evidence, as suggested by others, that endogenous host mechanisms impact the bacterial colonization of body sites of these obligate aquatic mammals and confirm that those mechanisms are resistant to major change in the in-contact water. Further work is likely to provide unique insight into the biological mechanisms of these important host–microbe relationships and inform management decisions toward optimizing animal health.

**Supplementary Materials:** The following supporting information can be downloaded at: https://www.mdpi.com/article/10.3390/oceans3030020/s1, Figure S1. Venn diagram illustrating unique and shared ASVs by site and source. Figure S2. Scatterplot tracking changes on salinity (top), richness (Observed ASVs—middle), and diversity (Faith's PD—bottom) in fecal community diversity. Points are colored based on salinity values. There were no significant changes in diversity. Figure S3.

Scatterplot tracking changes on salinity (top), richness (Observed ASVs—middle), and diversity (Faith's PD—bottom) in chuff community diversity. Points are colored based on salinity values. There were no significant changes in diversity. Figure S4. Barplot demonstrating the relative abundances of families within chuff samples throughout the water change. Salinities for each day are overlayed with black points. The 8 most abundant families across are samples are displayed in the legend. Figure S5. Barplot demonstrating the relative abundances of families within fecal samples throughout the water change. Salinities for each day are overlayed with black points. The 8 most abundant families across are samples are displayed in the legend. Figure S6. PCoA of the microbial communities from fecal samples throughout the water change. To aid visualization, samples are colored by phase opposed to salinity. There were no significant changes between phases. Figure S7. PCoA of the microbial communities from chuff samples throughout the water change. To aid visualization, samples are colored by phase opposed to salinity. There were no significant changes between phases. Table S1. PERMANOVA and PERMDISP results from comparisons between microbial communities from water and those associated with different whale body sites (skin, chuff, and feces) based on generalized UniFrac values. Significant results are bolded (*p*-adj. $< 0.05$). Table S2. Mean relative abundance and the standard error of the mean for the top 25 most abundant taxonomic phyla, families, and genera within water microbial communities from each of the four water change phases. Table S3. Mean relative abundance and the standard error of the mean for the top 25 most abundant taxonomic phyla, families, and genera within skin microbial communities from each of the four water change phases. Table S4. Mean relative abundance and the standard error of the mean for every taxonomic phyla, and the top 25 most abundant families, and genera within chuff microbial communities from each of the four water change phases. Table S5. Mean relative abundance and the standard error of the mean for every taxonomic phyla, and the top 25 most abundant families, and genera within fecal microbial communities from each of the four water change phases.

**Author Contributions:** Conceptualization, W.V.B. Methodology, W.V.B., L.P. and F.O.; formal analysis, L.P. and F.O.; resources, W.V.B.; data curation, L.P.; writing—original draft preparation W.V.B.; writing—review and editing, W.V.B., F.O. and L.P.; visualization, L.P.; project administration W.V.B.; funding acquisition, W.V.B. All authors have read and agreed to the published version of the manuscript.

**Funding:** This work was supported by the Institute of Museum and Library Services [National Leadership Grant MG-60-18-0018-18].

**Institutional Review Board Statement:** All animal use activities associated with this project were conducted under approval from the Shedd Aquarium Research Review Committee and in compliance with all Federal and State regulations. Protocol # 2020-03.

**Data Availability Statement:** All sequence reads were made available through the BioProject PRJNA680758 at the NCBI's Sequence Read Archive.

**Acknowledgments:** The authors thank Chrissy Gibbons for project management and coordination of sample collection with the Shedd Aquarium animal husbandry staff.

**Conflicts of Interest:** The authors declare no conflict of interest.

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
