# Peer review of "Whale-Associated Microbial Communities Remain Remarkably Stable despite Massive Water Community Disruption in a Managed Artificial Marine Environment"

_2673-1924, doi:10.3390/oceans3030020_

Round 1
Reviewer 1 Report
Whale and environmental microbial communities in an artificial marine environment
This paper is extremely well written, organized and provides valuable information in the impact of environmental change (salinity) on environmental and whale microbial communities. The tables and figures are well designed and informative and the references are pertinent to the work. The methods for the molecular analysis are detailed and replicate samples improved the validity of the results. Adequate controls were incorporated into the molecular analyses. The conclusions are supported by the data presented in the paper. There are a few minor suggestions and questions for clarification.
Line 38 with respect to the respiratory cycle being opposite to most terrestrial animals, that is they surface, exhale, inhale and dive. This statement is a confusing to me. For most terrestrial species, would they not also be breath holders when diving? Or, would the “opposite” reflect the premise that terrestrial mammals are initially at the surface, inhale, then dive?
Lines 58-60 The mention of habitat, environmental community and host with regards to microbial flora is well taken. However, as mentioned in the discussion wrt free ranging humpback whales and changes in profiles may change according to the health status of the animal. (Apprill et al, 2010). With the information provided in the introduction, paragraph 52-73, and in materials and methods, paragraph lines 108-118, there is little mention of information regarding the reproductive or health status of the animals during the study interval. Were there any changes apparent clinically (behavior, food intake, respiration rates, etc), were there any changes in the reproductive status of the animals or other health parameters which may have contributed to altering the enteric or cutaneous microbial flora? Would the authors consider adding some text with regards to microbial flora and competitive exclusion to prevent colonization and potential microbial invasion into tissues.
Lines 77-91, with regards to the environmental samples, the protocols are appropriate and well detailed in the text. Could the authors please comment on whether they may have considered collection of any water samples from the filtration system or equipment, would there have been any value in these?
Author Response
Thank you for the thoughtful review. We are confident the suggestions have improved the manuscript.

Reviewer 2 Report
This manuscript describes the microbial community in the water, dust, mist, and beluga whale microbiome during a water change. The manuscript concludes that the community of the water fluctuates during the different phases of the water change. However, there is no report on whether dust or mist changes over time and it's unclear why these samples were not further analyzed. The author's main conclusion is that the beluga microbiome is stable throughout the water change process, but only alpha diversity is tested. The authors need to conduct more statistical tests (PERMANOVA, dispersion analysis, differential abundance analysis) to make this claim. Overall the article needs to provide more statistical information, conduct more analysis, and develop the introduction a bit more. Below are my line-by-line comments.
Lines 25 -74 The introduction overall reads well but it's missing why this study is important. It would be informative if more background about the water change procedure is provided, such as how often it is conducted and why it’s conducted at this rate. Also, I think a clear goal of why it is important to evaluate the changes of the microbial community during water changes would give this research more substance. Some of these topics are covered in the discussion, but I think it is more appropriate to have it in the introduction to let the reader know why this research question matters. The introduction also needs to explain why collecting mist, and dust is valuable.
Line 157 it is unclear if the samples were normalized for both alpha and beta diversity analysis.
Line 185 -187 This reads like a figure legend instead of a sentence. Also, I think beta-diversity should have its own paragraph and the statistical values should be stated.
Line 217 -218 Report the dispersion values
Line 221- 224 Report the relative abundance changes.
Line 234-243 The authors conclude that there are no changes to the whale microbial community throughout the water change, but only alpha-diversity was used to address this question. Why didn’t the authors test beta-diversity among the different body sites through time and conduct a differential abundance analysis?
The headers throughout the manuscript should be more descriptive.
Author Response
Thank you for the thoughtful review and comments. We are confident the changes have addressed them and improved the manuscript.

Reviewer 3 Report
It is a unique study investigating marine mammal microbiomes during water exchange. I think this data are worth to publish. However, it would be good to take much more information out of the data and to present abundant genera and phyla and see if there are significant changes during the water exchange within specfic bacterial groups.
Comments:
L14: marine mammals? Plural?
Methods in general:
Please indicate if you have done rarefaction and if not, could you expain why you decided not to do that.
L151-162: Which database was used and which version? SILVA? There has been a change in the nomenclature of some phyla and also some re-organizations, e.g. within the proteobacteria. Has this been noticed?
Results in general:
You could remove two of the PCoA, as it is enough to show only one, maybe the one with the highest ‚expalantion percentages‘.
Could you please perform a Venn Diagram showing all ASVs that are shared or unique between environments and whales?
Barplots: Actually I also see some differences in the relative abundance of some families within one phase, for instance between Jan 08 and Jan 09. How can you explain that? I think it could be worth to test for significance for some abundant families or genera separately, e.g. by boxplots including statistical tests. This could show in more detail, what is stable and what changes with the different phases. And what is about individual differences of the animals. I assume the samples are pooled, but what about the variability among the individulas?
How do you explain the higher abundance of Vibrionaceae in the feces only at Feb 19 and Feb 26?
L219-220: Why did you decide to show only family level? It would also be interesting to show the most abundant genera and even phyla. Would it be possible to show the most abundant genera and phlya as well? You have the data and such information are missing. We do not know a lot about the microbiomes of marine mammals. Your data could help to fill the gap. In my opinion, it would be very worth mentioning. It could be shown as heatmap, table or barplots.
I would then also add the graphs for chuff and feces.
Discussion:
You should also discuss the introduction of new microbes via the new water. You have some higher ASVs abundances in the post phase in some cases (eg Vibrionacea in feces or uncultured Glacillibacteria in water).
L270-271: 48.744% relative abundance or 89.037% of the total community? This cannot be, as your barplots show no family that has such high percentages.
Author Response
Thank you for the thoughtful review and comments. We are confident the revisions have addressed them and improved the manuscript.

Round 2
Reviewer 1 Report
The revised manuscript is improved from the original version and read very well. I have not further edits or suggestions.
Author Response
Thank you for the input.
Reviewer 2 Report
I think the authors have really improved the revised manuscript with the updated figures and analysis. Although, the updated supplementary file was not made available to me, so I could not fully review the manuscript. In addition, I think the authors do not provide enough details in their methods section and this needs to be updated.
Line 14 Add gene after 16S rRNA
Line 174 What tools were used within phyloseq? What parameters were used? Was the data rarefied if so to what depth? This statement should be followed by how alpha-diversity was statistically tested. But it is mentioned later, and it is difficult to follow.
Line 191 Is the code of this analysis publically available? I don’t understand how BH correction was performed on a PERMANOVA with the programs listed was there another program used to conduct this? For both alpha and beta analysis, it should also state what factors are being tested. In line 181 the authors state “test for significant differences between communities“ which communities are being tested? This statement should include all tests conducted.
Line 192 why isn’t this mentioned in line 184?
Line 199 I think it would be informative to start the results with the number of samples per phase and the environmental source.
Line 202-205 Why is Shannon Diversity not mentioned in the results? There are also other missing results. Other alpha-diversity pairwise comparisons were significant but not mentioned in the results.
Line 247 This sort of information should be stated in the methods to let the reader know that continuous data were analyzed as categorical (pre, transition, hold, and post) to test for alpha-diversity.
Line 312 This was not the case for Shannon diversity. Please add to the discussion.
Line 376 This result may also be from a short-term environmental challenge, and prolonged stress may yield different results.
Line 388 This can be true, but to some extent, since marine mammals can develop lesions when exposed to poor water quality.
Line 414 I did not find the updated supplementary file and can only find the original supplementary file.
Author Response
Thank you again for the insightful comments and suggestions. Specific responses are in the attached document.

Reviewer 3 Report
In your responses, please give line numbers of the text you added in the manuscript, so that the reviewer can easily follow and understand your changes.
Otherwise the reviewer need to look up the whole text and search for the changed text and the time of a reviewer is limited.
Lines 211-217: Be consistent with the form of number (comma or not).
Line 212: remove ‘were’
I cannot see Figure S1 (Venn diagram) in your Supplements. It is missing or not the updated version. The same for Table S2-S5. I cannot find them.
Lines 332-334: I still don’t understand these percentages. In Figure 8 the families Flavobacteriaceae and Moraxellaceae do not even reach 25%. How can a genus like Psychrobacter, which is a member of the Moraxellaceae be 89%?
Figure 8: Did you test for any significance? You said that the whale skin microbiome stays stable, but in Figure 8 you can see families that differ significantly between the phases, e.g. Saccharospirillaceae are much less in the post phase in comparison to the pre and transition phase. Please be more precise, why you state that the microbiome is stable and where there are changes or variation.
Author Response
Thank you again for the thoughtful comments and suggestions. Specific responses are included in the attached document.

Round 3
Reviewer 2 Report
My only suggestion is for the authors to add legends to the supplementary tables.
Reviewer 3 Report
It can be accepted now.